# The Use of Bruton Tyrosine Kinase Inhibitors in Waldenström’s Macroglobulinemia

**DOI:** 10.3390/jpm12050676

**Published:** 2022-04-22

**Authors:** Abdullah Mohammad Khan

**Affiliations:** Division of Hematology, Department of Internal Medicine, The Ohio State University, Columbus, OH 43210, USA; Abdullah.Khan@osumc.edu

**Keywords:** Waldenström’s macroglobulinemia, Bruton tyrosine kinase inhibitor, ibrutinib, acalabrutinib, zanubrutinib

## Abstract

Waldenström’s macroglobulinemia (WM) remains an incurable malignancy. However, a number of treatment options exist for patients with WM, including alkylating agents, anti-CD20 monoclonal antibodies, and small molecule inhibitors such as proteasome inhibitors and Bruton tyrosine kinase inhibitors (BTKi). The focus of this review is to highlight the role of BTKi in the management of WM. The first BTKi to receive US Food and Drug Administration approval for WM was ibrutinib. Ibrutinib has been extensively studied in both treatment-naïve WM patients and in those with relapsed/refractory disease. The next BTKi approved for use was zanubrutinib, and prospective data for acalabrutinib and tirabrutinib have also recently been published. Efficacy data for BTKi will be discussed, as well as the differences in their adverse event profiles.

## 1. Introduction

Waldenström’s macroglobulinemia (WM) is an indolent B-cell lymphoproliferative disorder characterized by bone marrow infiltration with lymphoplasmacytic cells in association with a monoclonal immunoglobulin M (IgM) gammopathy [1]. It is a relatively uncommon non-Hodgkin lymphoma (NHL) with an incidence rate of 0.3 per 100,000 persons [2]. WM is a disease of the elderly, with a predilection for males more than females, and for Caucasians more than other populations [3]. There may be genetic and environmental factors which contribute to the development of WM, but no clear causative factors have been identified in the pathogenesis [4]. Based on a recent Surveillance, Epidemiology, and End Results (SEER) analysis, the five-year survival rate has improved from 48% to 69% during the period from the 1980s to the 2010s [3].

WM patients typically present with symptoms either related to disease infiltration into hematopoietic tissues or related to macroglobulinemia. The National Comprehensive Cancer Network (NCCN) guidelines suggest the following as essential workup of patients with WM: history and physical exam, complete blood count, renal and hepatic function tests, serum calcium, uric acid, lactate dehydrogenase, beta-2 microglobulin, quantitative immunoglobulins, protein electrophoresis, immunofixation, CT scan of chest/abdomen/pelvis, bone marrow aspirate/biopsy, and *MYD88^L265P^* analysis. Pertinent to the topic of this review, *CXCR4* mutational analysis is also recommended.

The Second International Workshop on WM detailed indications for treatment based on symptomatology and labs [5]. Symptoms that should prompt treatment include B symptoms (recurrent fever, night sweats, weight loss, and fatigue), symptoms related to hyperviscosity (classic triad of mucosal bleeding and visual and/or neurologic disturbances), sensorimotor peripheral neuropathy, symptomatic lymphadenopathy or splenomegaly, systemic amyloidosis, and cryoglobulinemia. Progression of anemia (hemoglobin ≤ 10g/dL) or thrombocytopenia (platelets < 100 × 10^9^/L) attributable to WM is also an indication for treatment. WM must be differentiated from IgM monoclonal gammopathy, multiple myeloma, and other lymphomas. For example, IgM multiple myeloma is a rare disorder but may have translocation t(11;14) or lytic lesions that are classically not seen in WM. t(11;14) can also be seen in mantle cell lymphoma, as can nuclear staining for cyclin D1, but this is not seen in WM.

Traditionally, treatment guidelines, including those presented in the Second International Workshop and in earlier versions of the NCCN guidelines, preferred alkylating agents, nucleoside analogues, and rituximab as first-line treatments for WM. However, groundbreaking research has delineated *MYD88* and *CXCR4* genes in the pathogenesis of WM and identified the Bruton tyrosine kinase (BTK) as a potent therapeutic target. Treon et al. performed whole-genome and Sanger sequencing in patients meeting the diagnostic criteria for WM, and identified *MYD88^L265P^* as the most common somatic alteration in the lymphoplasmacytic cells [6,7]. It is worth noting *MYD88^L265P^* is not specific to WM; a recent paper demonstrated the presence of *MYD88^L265P^* in normal precursor and mature B lymphocytes from patients with lymphoma [8]. The authors concluded *MYD88^L265P^* is a preneoplastic event and that additional genetic changes are required for lymphomagenesis. MYD88 is an adaptor molecule for Toll-like receptors. Following receptor stimulation, the MYD88 homodimer forms complexes with IRAK4 (the Myddosome), leading to activation of IL-1 receptor associated kinase 1 (IRAK1) and IRAK2 [9]. IRAK1 activates tumor necrosis factor receptor-associated factor 6 (TRAF6) and causes phosphorylation of IκBα, which then activates nuclear factor κβ (NFκβ) survival signaling. *MYD^mut^* results in Myddosome self-assembly and can activate NFκβ without receptor signaling through IRAK1/IRAK4 or BTK [10]. BTK is a downstream component of *MYD88^L265P^* signaling in WM, and treatment with BTKi results in inhibition of IκBα phosphorylation and, thus, blocks NFκβ activation. *MYD88^L265P^* WM also leads to expression and activation of another pro-survival kinase, hematopoietic cell kinase (HCK) [10]. BTKi bind to the ATP-binding pocket of HCK and block HCK activity.

Clinically relevant somatic mutations are also found in the C-terminal domain of the C-X-C chemokine receptor type 4 (*CXCR4*) gene in ~30% of WM patients [11]. CXCL12 binding to CXCR4 promotes survival and migration of WM cells to the BM. *CXCR^mut^* results in loss of regulatory serines that would typically prevent sustained CXCL12-mediated activation of the AKT and extracellular regulated kinase (ERK) pathways [12]. Because of this enhanced AKT/ERK signaling, conventional WM therapy and BTKi have less efficacy in *CXCR*^mut^ WM cells [13].

In addition to impacting pathogenesis, the presence or absence of these mutations has an impact on the clinical manifestation of WM [14]. Patients with *MYD88^WT^*/*CXCR4^WT^* demonstrated the least bone marrow involvement, and the lowest serum IgM concentration, compared to other genotypes. On the other hand, *MYD88^L265P^*/*CXCR4^WT^* patients had a higher prevalence of adenopathy and symptomatic disease compared to other genotypes. Nonsense mutations in *CXCR4* were also observed more frequently in WM patients with low levels of von Willebrand markers compared to normal markers [15]. Finally, *MYD88^WT^* is also associated with a higher risk of transformation to a high-grade lymphoma compared to *MYD88^L265P^* [16]. As will be discussed later, there are also important differences in BTKi activity depending on the WM genotype.

## 2. Ibrutinib

Ibrutinib is a selective and irreversible BTKi which blocks BCR signaling downstream of BTK [17]. It also has the off-target effect of inhibiting HCK. In a phase I trial of ibrutinib in patients with relapsed/refractory (R/R) NHL, three out of four WM patients demonstrated a partial response (PR) and the fourth had stable disease (SD) [18]. Long-term follow-up of the pivotal phase II multicenter study of ibrutinib in R/R WM patients demonstrated an overall response rate (ORR; ≥minimal response, MR) of 90.5%, and a major response rate (MRR; ≥PR) of 79.4% [19,20]. No complete responses (CR) were observed, but the rate of very good partial responses (VGPR) increased over time to 30.2%. Responses were impacted by *MYD88* and *CXCR4* mutation status. *MYD88^mut^*/*CXCR4^mut^* were associated with lower major responses than *MYD88^mut^*/*CXCR4^WT^* (68.2% vs. 97.2%, respectively) and VGPR rates (47.2% vs. 9.1%, respectively). Furthermore, the median time to major response was longer in *MYD88^mut^*/*CXCR4^mut^* patients compared to *MYD88^mut^*/*CXCR4^WT^* patients (4.7 months vs. 1.8 months, respectively). As predicted by the disease pathobiology, *MYD88^WT^* patients (*n* = 4) did not have any major responses. Finally, the median PFS was not reached for patients with *MYD88^mut^*/*CXCR4^WT^*, it was 4.5 years for *MYD88^mut^*/*CXCR4^mut^*, and 0.4 years for *MYD88^WT^* patients. Based on the initial publication, ibrutinib was approved by the US Food and Drug Administration to treat symptomatic WM.

The efficacy of ibrutinib monotherapy was specifically evaluated in the treatment-naïve (TN) WM population [21]. Similar to above, excellent ORR (100%) and MRR (83%) were observed in totality, but the quality and time to major responses varied by genotype. Of note, the median time to major response was 1.8 months in the *MYD88^mut^*/*CXCR4^WT^* group, compared to 7.3 months in the *MYD88^mut^*/*CXCR4^mut^* group. The estimated 18-month PFS was 92%, and all patients were alive at the time of publication.

An open-label sub-study of the phase III iNNOVATE trial evaluated the efficacy of ibrutinib in patients who failed to achieve at least a minor response or relapsed <12 months after their last rituximab-containing therapy [22]. In the final analysis, at a median follow-up of 58 months, the median PFS was 39 months, and median OS was not reached [23]. When comparing the different genotypes, the median PFS in the *MYD88^L265P^*/*CXCR^mut^* and *MYD88^L265P^*/*CXCR4^WT^* groups was 18 months and not reached, respectively. As with the previous study, VGPR rates were higher in the *MYD88^L265P^*/*CXCR4^WT^* group compared to the *MYD88^L265P^*/*CXCR^mut^* group (41% vs. 14%), and the median time to major response was shorter in the former group compared to the latter (1 month vs. 4 months). Median OS rates were not reached, 50 months, and 9 months, for *MYD88^L265P^*/*CXCR4^WT^*, *MYD88^L265P^*/*CXCR^mut^*, and *MYD88^WT^*, respectively.

Acquired mutations associated with ibrutinib resistance have been observed. In a small study of six patients with progressive disease, half had BTK C481S mutations [24]. The mechanism of ibrutinib resistance appears to be through a paracrine process whereby reactivation of ERK1/2 signaling is accompanied by the release of pro-survival and inflammatory cytokines [25]. Targeted deep next-generation sequencing also identified mutations in phospholipase C ɣ 2 (PLCɣ2) and caspase recruitment domain family member 11 (CARD11); these mutations have also been observed in ibrutinib-resistant patients in other disease states [24]. While the role of mutations in PLCɣ2 has been inferred to result in aberrant signaling, based on observations in other NHL, specific functional studies are needed to determine the role of these mutations in patients with WM. Finally, genomic evolution has also been observed in patients relapsing during ibrutinib therapy. Whole exome sequencing of five patients with progressive disease detected progressive expansion of tumor clones with deletions of 6q and 8p [26]. These regions include genes important in regulating BTK, MYD88, and NFκβ signaling, as well as apoptotic pathways.

## 3. Acalabrutinib

Acalabrutinib is a second-generation BTKi that contains a reactive butynamide group which covalently binds to Cys481 in BTK [27]. It has a higher selectivity for BTK compared to ibrutinib, and has fewer off-target actions related to the epidermal growth factor receptor (EGFR) and interleukin 2-inducible T cell kinase (ITK). Acalabrutinib efficacy was studied in 14 patients with TN WM, and 92 patients with R/R disease [28]. After a median follow-up of 27.4 months, the ORR for both TN and R/R patients was 93%, and the MRR was 79% versus 80%, respectively. Although the patients’ *MYD8^mut^* status was known, their *CXCR^mut^* status was not. VGPR were not observed in the *MYD88^WT^* subset and they were 28% in the *MYD88^L265P^* subset. This contributed to the observed differences in ORR (94% vs. 79%) and MRR (78% vs. 57%) between *MYD88^L265P^* and *MYD88^WT^* patients. The most common grade 3–4 events were neutropenia and pneumonia. Grade 3–4 atrial fibrillation occurred in one patient, and grade 3–4 bleeding occurred in three patients.

## 4. Zanubrutinib

Zanubrutinib is another next-generation BTKi which covalently binds to Cys481 that, compared to ibrutinib, demonstrates greater selectivity for BTK and demonstrates fewer off-target effects [29]. Zanubrutinib also possesses better selectivity against EGFR, TEC, and Src family kinases, compared to ibrutinib [30]. In the phase I study of zanubrutinib in R/R B-cell malignancies, sustained peripheral blood mononuclear cells (PBMCs) and nodal BTK occupancy was observed at the recommended phase 2 dose (RP2D) of 160 mg twice daily by mouth [29]. In the phase III ASPEN trial, TN and R/R WM patients were randomized to receive ibrutinib or zanubrutinib [31]. In the TN patients, ORR and MRR were comparable between ibrutinib and zanubrutinib (89% vs. 95%, and 67% vs. 74%, respectively). The investigators indicated there was a trend towards deeper levels of remission in the zanubrutinib group (≥VGPR: 17% vs. 26%), but progression rates were numerically superior with ibrutinib (18-month EFS: 94% vs. 78%). In this author’s review of the data, there were no obvious differences in baseline disease characteristics, including *CXCR4^mut^* status, that would help account for this difference in outcome. In the R/R setting, in patients with a median of one prior line of therapy, similar ORR and MRR were observed in the two groups (ibrutinib: 94% and 80%, respectively; zanubrutinib: 94% and 78%, respectively). As in the TN setting, rates of VGPR or better demonstrated a trend towards deeper remission with zanubrutinib, but did not reach clinical significance (20% vs. 29%; *P* = 0.12). With regards to safety, atrial fibrillation, diarrhea, contusion, muscle spasms, peripheral edema, and pneumonia were noted more frequently in patients treated with ibrutinib, and neutropenia was noted more frequently in patients treated with zanubrutinib.

Zanubrutinib has also appears to have better efficacy than other BTKi in patients with WM lacking activating mutations in the *MYD88* gene (i.e., *MYD88^WT^*). In a sub-study of the patients in the ASPEN trial, 23 patients had R/R disease and 5 patients were TN [32]. The 18-month PFS was 68% and OS was 88%. Of interest, ≥VGPR was seen in 29% of patients with R/R disease, MRR was 52%, and ORR was 81%. Finally, clinical activity with zanubrutinib was also observed specifically in the Asian R/R WM patient population in a phase II study [33].

## 5. Other BTKi and Combination Studies

Other BTKi are also in development. Tirabrutinib demonstrated impressive ORR and MRR in both TN and R/R Japanese WM patients [34]. In the updated analysis with a median follow-up of approximately two years, the ORR and MRR were 94% in TN patients, and 100% and 89%, respectively, in R/R patients [35]. VGPR or better was observed in 41% of patients with *MYD88^L265P^*/*CXCR4^WT^*, and was not observed in the other two genotypes, although the number of patients was limited. The MRR per genotype was 91% for *MYD88^L265P^*/*CXCR4^WT^*, and 100% for both *MYD88^L265P^*/*CXCR^mut^* and *MYD88^WT^*. All patients were alive at the time of data cutoff; the PFS rate at 24 months in the TN group was 94%, and 89% in the R/R group. Grade ≥ 3 AEs attributed to the study drug included neutropenia, leukopenia, lymphopenia, hypertriglyceridemia, retinal detachment, atypical mycobacterial infection, and skin-related AEs.

A large phase III study attempted to elucidate the role of ibrutinib and rituximab combination therapy in patients with treatment-naïve and R/R WM [36]. Final analysis of the iNNOVATE study demonstrated a number of benefits in combining BTK with CD20 inhibition [37]. Although there were PFS (median NR placebo–rituximab arm vs. 20 months ibrutinib–rituximab arm) and ORR (92% placebo–rituximab arm vs. 44% ibrutinib–rituximab arm) benefits for the combination therapy, there were no differences in OS. The 54-month PFS rates by genotype analysis in the combination therapy and rituximab arms were 72% versus 25% in the *MYD88^L265P^*/*CXCR4^WT^* group, 63% versus 21% in the *MYD88^L265P^*/*CXCR^mut^* group, and 70% versus 30% in the *MYD88^WT^* group, respectively. Interestingly, the improvements in ORR were seen regardless of genotype. Investigators also noted a decreased incidence of infusion reactions (43% vs. 59%) and IgM flare (8% vs. 47%) with the combination, compared to single-agent rituximab. A common criticism of the study is that it did not include a single-agent ibrutinib comparator arm; in chronic lymphocytic leukemia (CLL), there are data that indicate no significant difference between ibrutinib and ibrutinib plus rituximab, with regards to PFS [38]. Efficacy data for the discussed trials is presented in Table 1.

## 6. Discussion

Despite rapid drug development, there is no cure for WM, so a number of factors must be reviewed when considering treatment options. First of all, the goal of treatment must be to control symptoms and to prevent or reverse end-organ damage, as well as to improve the quality of life. Secondly, treatment must be tailored to patient factors such as age, frailty, comorbid conditions, and patient preferences. Treatment must also take into consideration disease-related factors such as emergent disease manifestations of hyperviscosity, presence or absence of other WM symptoms, and disease genotype data.

For many providers, the standard-of-care for most young and fit patients in the TN setting remains a combination of an alkylator such as bendamustine, and rituximab (BR). This is based on a phase III Study Group Indolent Lymphomas trial that compared BR to R-CHOP (cyclophosphamide, doxorubicin, vincristine, and prednisone) [39,40]. With six cycles of therapy, BR therapy resulted in a median PFS of 69.5 months, versus 28.1 months in the R-CHOP group, while also resulting in fewer serious AE. Although cross-trial comparisons are not ideal, single-agent ibrutinib results in an 18-month PFS exceeding 90%. Furthermore, the 48-month PFS of ibrutinib–rituximab in TN patients was 70%. Thus, for less fit patients with the more common *MYD88^L265P^*/*CXCR4^WT^*, either combination ibrutinib–rituximab or single-agent ibrutinib are appropriate treatment options. However, the decision making becomes more challenging when considering the smaller subset of TN WM patients with either *MYD88^WT^* or *MYD88^L265P^*/*CXCR4^mut^*, as prospective data is limited. A retrospective analysis of French WM patients treated with BR showed that although disease response was not impacted by *MYD88* or *CXCR4* mutation status, PFS was shorter in patients with *MYD88^WT^*, but not in those with *CXCR4^mut^* [41].

For patients with R/R disease who did not have frontline BTKi therapy, or who had a short duration of remission with alkylator-based therapy, BTKi are an attractive treatment option, given the efficacy and safety data. BTKi therapy in R/R WM with *MYD88^L265P^*/*CXCR4^WT^* genotype yields >90% response rates and, in the case of single-agent ibrutinib, a 60-month overall survival rate of 87%. By comparison, acalabrutinib demonstrated a 24-month OS rate of 89%, and zanubrutinib demonstrated an 18-month OS rate of ~90%. In this patient population, since efficacy data appears similar, choice of BTKi can be based on safety profile and availability (Table 2). All BTKi may result in cytopenias, but zanubrutinib and tirabrutinib appear to have a higher incidence of high-grade neutropenia. All of these BTKi appear to increase the risk of atrial fibrillation and hypertension, but high-grade manifestations may be more common with ibrutinib compared to the other two. Similarly, grade 3–4 hemorrhage appears to be more common with ibrutinib. Finally, headache is a unique side effect reported more frequently in patients treated with acalabrutinib compared to the other drugs, being observed in 39% of acalabrutinib-treated patients [28].

Once again, for patients with R/R disease and either *MYD88^WT^* or *MYD88^L265P^*/*CXCR4^mut^*, treatment outcomes are inferior compared to the other genotypes. Based on inferior response rates with ibrutinib monotherapy, the NCCN recommends consideration of *CXCR4* mutation testing when starting therapy. Based on the above data, ibrutinib–rituximab or zanubrutinib may be more appropriate if considering BTKi therapy for this patient population. Clinical trial participation should be encouraged. For example, next-generation BTKi that are designed to inhibit both WT and C481S mutated BTK, such as pirtobrutinib, may have a role in helping these patients [42]. However, thought-provoking in vitro data showed that having both a C481S mutated BTK and certain gatekeeper mutations in the BTK catalytic domain resulted in “super-resistance” to acalabrutinib and zanubrutinib, in addition to ibrutinib [43]. It is not clear if these gatekeeper mutations are observed in ibrutinib-refractory patients, but this offers insight into mechanisms of resistance. Furthermore, as discussed before, genomic evolution is also observed in ibrutinib-refractory patients, and re-challenging with a next-generation BTKi may not demonstrate clinical efficacy.

A number of other small molecule inhibitors and antibody therapies are in development and may have a role to play in patients with *MYD88^WT^*, *MYD88^L265P^*/*CXCR4^mut^*, or disease refractory to BTKi. Combination studies with venetoclax are exciting since venetoclax monotherapy resulted in an ORR of 84% and MRR of 81%, and treatment responsiveness was not affected by *CXCR4^mut^* [44]. As previously discussed, *CXCR4^mut^* is associated with inferior responses to ibrutinib compared to *CXCR4^WT^*, partly due to enhanced activation of CXCR4^mut^ by its ligand CXCL12. Therefore, blocking this interaction may increase the clinical activity of BTKi. Ulocuplumab is a first-in-class monoclonal antibody that binds to CXCR4 and blocks CXCL12 binding to it [45,46]. Investigators examined the combination of ibrutinib and ulocuplumab in BTKi-naïve WM patients harboring the *CXCR4^mut^*. The ORR and MRR was 100%, and 4 out of 12 patients attained a VGPR. Responses were rapid (median time to major response was 1.2 months), and therapy was well tolerated. Common grade ≥ 2 AE included thrombocytopenia, rash, skin infection, fatigue, diarrhea, cough, and hyperglycemia. Despite safety and efficacy data, the sponsor terminated development of ulocuplumab, but other CXCR4 antagonists are in development [47].

Nonetheless, despite the impressive treatment responses seen with BTKi, in the author’s opinion they do not represent the ideal treatment option, due to the indefinite duration of therapy. Akin to CLL and multiple myeloma therapy, combination approaches may provide deeper responses and, for patients attaining minimal residual disease status, provide an opportunity to cease therapy and commence a period of active surveillance. Alternatively, a response-adaptive approach may be needed for patients not attaining at least a PR to therapy. A recent study showed three-year PFS rates for WM patients undergoing ibrutinib monotherapy who attained major responses, versus those who did not, were 83% and 54%, respectively [47].

## 7. Conclusions

WM is a rare NHL, and conducting large, prospective clinical trials for these patients is difficult without multi-institutional efforts. BTKi represent a fascinating class of therapeutics stemming from ground-breaking discoveries in the basic science of B-cell malignancies. BTKi have been shown to be safe and highly effective in patients with WM. With further understanding of the biology of WM and the mechanisms of treatment resistance, it is likely that outcomes for all patients, regardless of genotype, can be improved. Single-agent and combination BTKi studies are ongoing and/or are planned to further elucidate the role of BTKi in the care of patients with WM.

## Figures and Tables

**Table 1 jpm-12-00676-t001:** Efficacy outcomes in large BTKi trials in TN and R/R WM.

Publication	Trial Phase	Therapy	Setting	Median Prior Therapies (Range)	*N*	ORR	MRR	PFS	OS
Treon, 2018	II	Ibrutinib	TN	N/A	31	100%	83%	18 m—92%	*
Buske, 2022	III	Ibrutinib + Rituximab	TN	N/A	34	91%	76%	48 m—70%	NR
		Placebo + Rituximab	TN	N/A	34	41%	41%	48 m—32%	NR
Owen, 2020	II	Acalabrutinib	TN	N/A	14	93%	79%	24 m—90%	24 m—92%
Tan, 2020	III	Ibrutinib	TN	N/A	18	89%	67%	18 m—94%	18 m—93% **
		Zanubrutinib	TN	N/A	19	95%	74%	18 m—78%	18 m—97% **
Dimopoulos, 2020	III	Zanubrutinib	TN	N/A	5	80%	40%	18 m—60%	18 m—80%
Sekiguchi, 2022	II	Tirabrutinib	TN	N/A	18	94%	94%	24 m—94%	24 m—100%
Treon, 2021	II	Ibrutinib	R/R	2 (1–9)	63	90%	79%	60 m—54%	60 m—87%
Buske, 2022	III	Ibrutinib + Rituximab	R/R	2 (1–6)	41	93%	76%	48 m—71%	NR
		Placebo + Rituximab	R/R	2 (1–6)	41	53%	22%	48 m—20%	NR
Trotman, 2021	III	Ibrutinib	R/R	4 (1–7)	31	87%	77%	39 m	NR
Owen, 2020	II	Acalabrutinib	R/R	2 (1–7)	92	93%	78%	24 m—82%	24 m—89%
Tan, 2020	III	Ibrutinib	R/R	1 (1–6)	81	94%	80%	18 m—82%	18 m—93% **
		Zanubrutinib	R/R	1 (1–8)	83	94%	78%	18 m—86%	18 m—97% **
An, 2021	III	Zanubrutinib	R/R	2 (1–6)	44	77%	70%	24 m—60%	NR
Dimopoulos, 2020	III	Zanubrutinib	R/R	1 (1–5)	21	81%	52%	18 m—68%	18 m—88%
Sekiguchi, 2022	II	Tirabrutinib	R/R	***	9	100%	89%	24 m—89%	24 m—100%
Mato, 2021	I/II	Pirtobrutinib	R/R	3 (2–4)	26	68%	47%	NR	NR

Abbreviations: N—number; ORR—Overall response rates; MRR—Major response rates; PFS—Progression free survival; OS—Overall survival; TN—Treatment naïve; R/R—Relapsed/refractory; N/A—Not applicable; NR—Not reported; m—Months. * No deaths were reported at median follow-up of 14.6 months. ** Percentages included TN and R/R patients. *** In the initial manuscript, median prior lines of therapy were 2, and range was 1–7.

**Table 2 jpm-12-00676-t002:** Incidence of grade 3–4 adverse events of interest for BTKi in WM.

Adverse Events	Treon, 2021	Buske, 2022	Owen, 2020	Sekuguchi, 2022	Tan, 2020	
	Ibrutinib	Ibrutinib + Rituximab	Acalabrutinib	Tirabrutinib	Ibrutinib	Zanubrutinib
	*N* = 63	*N* = 75	*N* = 106	*N* = 27	*N* = 98	*N* = 101
Anemia	1%	12%	5%	Not reported	5%	5%
Thrombocytopenia	7%	1%	4%	0%	3%	6%
Neutropenia	10%	13%	16%	22%	8%	20%
Atrial fibrillation	1%	16%	1%	0%	4%	0%
Hypertension	0%	15%	3%	0%	11%	6%
Nausea/vomiting	0%	Not reported	3%	0%	2%	0%
Diarrhea	0%	Not reported	2%	0%	1%	3%
Upper respiratory tract infections	0%	Not reported	0%	0%	1%	0%
Pneumonia	2%	11%	7%	0%	7%	1%
Arthralgia/myalgia	0%	4%	3%	0%	1%	8%
Hemorrhage/bleed	0%	7%	3%	0%	9%	6%

## Data Availability

Not applicable.

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
