# Peer review of "The Use of Bruton Tyrosine Kinase Inhibitors in Waldenström’s Macroglobulinemia"

_jpm, 2022, doi:10.3390/jpm12050676_

Round 1

Reviewer 1 Report

Dr. Abdullah M. Khan reviews the current status of the use of Bruton tyrosine-kinase inhibitors (BTKi) for the management of Waldenström’s macroglobulinemia (WM), by performing a syntheses of different clinical trials in which this therapy is implemented. Although this year some reviews have been published with updated information regarding BTKi for the treatment of WM, such as: doi: 10.3389/fonc.2021.801124. I consider this manuscript is interesting since it is focused only in BTKi, providing more detailed information about adverse events and the results of different clinical trials. The manuscript is well written and readable for experts and non-experts in the fields of WM or BTKi. However, I have some suggestions and revisions throughout the text:

Major revisions:

  • Regarding the different mechanisms of resistance to BTKi in the field of WM, there are two recent works:

-Jiménez C, Chan GG, Xu L, Tsakmaklis N, Kofides A, Demos MG, Chen J, Liu X, Munshi M, Yang G, Castillo JJ, Wiestner A, García-Sanz R, Treon SP, Hunter ZR. Genomic evolution of ibrutinib-resistant clones in Waldenström macroglobulinaemia. Br J Haematol. 2020 Jun;189(6):1165-1170. doi: 10.1111/bjh.16463. Epub 2020 Feb 27. PMID: 32103491; PMCID: PMC7299825

- Piazza F, Di Paolo V, Scapinello G, Manni S, Trentin L and Quintieri L (2022) Determinants of Drug Resistance in B Cell-Non-Hodgkin Lymphomas: The Case of Lymphoplasmacytic Lymphoma/ Waldenström Macroglobulinemia. Front. Oncol. 11:801124. doi: 10.3389/fonc.2021.801124.

I suggest that these works be considered when addressing the resistance mechanisms, since they can further enrich the current work.

  • The legend of Table 2 is missing. Also, bearing in mind that this table refers to general considerations of BTKi, it should be included in sections prior to the discussion. For example, Table 2 could be mentioned as part of the introduction and in the particular sections of the inhibitors.
  • I suggest creating a specific section for Tirabrutinib, which includes the recent update of Sekiguchi N, et al. 2022 (https://doi.org/10.1111/cas.15344)
  • I suggest to mentioned the updates of Zabrutinib in the corresponding section (doi: 10.1182/bloodadvances.2021005621)
  • I would appreciate some comment on potential mechanisms of resistance to BTKi other than Ibrutinib.

 Minor revisions:

  • In the abstract, it is not necessary to indicate (TN) and (R/R) (line 12 and line 13, respectively).
  • In the introduction, 109 should be replaced by 109 (line 45) and CXCRmut should be replaced by CXCRmut (line 75).
  • In the “Ibrutinib” section, the meaning of R/R should be indicated (line 89). Also, considering the advance on
  • In the “Acalabrutinib” section, the meaning of TN should be indicated (line 131).
  • In the “Zanubrutinib” section, 160mg should be replaced by 160 mg (line 146).
  • In the “Other BTKi and Combination Studies” section, the meaning of AE should be indicated (line 177) and treatment-naïve should be replaced by TN
    (line 180).
  • In the discussion, I suggest to rewrite the sentence: “Despite rapid drug development, there is no cure for WM so a number of factors must be considered when considering treatment options” (line 204-205), because the words “considered” and “considering” sound repetitive to me.
  • In the discussion, capital letter should be removed in the word treatment (line 207).

Author Response

I thank the reviewer for their insightful comments and suggested references. 

Major revisions:

1) Added to the section on mechanisms of ibrutinib resistance including the first suggested reference (the second reviewed published data). Furthermore, in discussion section, separated paragraph describing next generation BTKi so there is additional commentary on role of next-gen BTKi after ibrutinib and upcoming therapies. Have added reference that comments on potential mechanisms of resistance with acalabrutinib and zanubrutinib. 

2) Appreciate comments regarding Table 2. Rather than add legend with section for abbreviations expanded them in the table (ex. URTI changed to upper respiratory tract infection). My preference is to leave the location of Table 2 as is. I wanted to draw the attention of the reader to the fact that since efficacy data is comparable between the different BTKi, side effect profile can be reviewed in one place of the discussion.  Tirabrutinib was added to Table 2 and added to the preceding discussion of side effect profiles. While I have reviewed the  suggested zanubrutinib manuscript, my preference is to restrict the reviewed articles to those focused on WM and not other B-cell malignancies. 

Minor revisions: 

  • In the abstract, it is not necessary to indicate (TN) and (R/R) (line 12 and line 13, respectively) --- removed [also removed FDA abbreviation]
  • In the introduction, 109 should be replaced by 109 (line 45)  and CXCRmut should be replaced by CXCRmut (line 75). --- done
  • In the “Ibrutinib” section, the meaning of R/R should be indicated (line 89). Also, considering the advance on --- done, also expanded on mechanisms of resistance
  • In the “Acalabrutinib” section, the meaning of TN should be indicated (line 131). --- added to preceding ibrutinib section when first mentioned
  • In the “Zanubrutinib” section, 160mg should be replaced by 160 mg (line 146).
  • In the “Other BTKi and Combination Studies” section, the meaning of AE should be indicated (line 177) and treatment-naïve should be replaced by TN
    (line 180). --- done
  • In the discussion, I suggest to rewrite the sentence: “Despite rapid drug development, there is no cure for WM so a number of factors must be considered when considering treatment options” (line 204-205), because the words “considered” and “considering” sound repetitive to me. --- changed first to "reviewed"
  • In the discussion, capital letter should be removed in the word treatment (line 207).

Reviewer 2 Report

Consider adding into the introduction aspects to differentiate WM form other heme malignancies, for instance cyclin D1 and bone lesions for MM vs WM.

Well organized and referenced background on the molecular characterization of the disease.

Nicely written. Have no other substantive comments

Author Response

I thank the reviewer for their insight. Have added to the 3rd paragraph of the introduction aspects to differentiate from other hematologic malignancies.  

Round 2

Reviewer 1 Report

All my suggestions were answeared. I consider that the work has been improved.Congratulations to the author for the nice work.